# Long-Term Effect of a Gluten-Free Diet on Diarrhoea- or Bloating-Predominant Functional Bowel Disease: Role of the ‘Low-Grade Coeliac Score’ and the ‘Coeliac Lymphogram’ in the Response Rate to the Diet

**DOI:** 10.3390/nu13061812

**Published:** 2021-05-26

**Authors:** Fernando Fernández-Bañares, Beatriz Arau, Agnès Raga, Montserrat Aceituno, Eva Tristán, Anna Carrasco, Laura Ruiz, Albert Martín-Cardona, Pablo Ruiz-Ramírez, Maria Esteve

**Affiliations:** 1Department of Gastroenterology, Hospital Universitari Mutua Terrassa, 08221 Terrassa, Spain; beatrizarau@mutuaterrassa.es (B.A.); araga@mutuaterrassa.cat (A.R.); maceituno@mutuaterrassa.es (M.A.); etristan@mutuaterrassa.cat (E.T.); anna.carrasco.garcia@gmail.com (A.C.); lruiz@mutuaterrassa.es (L.R.); albertmartin@mutuaterrassa.cat (A.M.-C.); pruiz@mutuaterrassa.es (P.R.-R.); mariaesteve@mutuaterrassa.cat (M.E.); 2Centro de Investigación Biomédica en Red de Enfermedades Hepáticas y Digestivas (CIBERehd), Instituto de Salud Carlos III, 28029 Madrid, Spain

**Keywords:** functional bowel disease, gluten-free diet, coeliac disease, tissue biomarkers, non-coeliac gluten sensitivity, FODMAP diet

## Abstract

1. Background: The long-term effect of a gluten-free diet (GFD) on functional bowel disorders (FBDs) has been scarcely studied. The aim was to assess the effect of a GFD on FBD patients, and to assess the role of both the low-grade coeliac score and coeliac lymphogram in the probability of response to a GFD. 2. Methods: 116 adult patients with either predominant diarrhoea or abdominal bloating, fulfilling Rome IV criteria of FBD, were treated with a GFD. Duodenum biopsies were performed for both pathology studies and intraepithelial lymphocyte subpopulation patterns. Coeliac lymphogram was defined as an increase in TCRγδ^+^ cells plus a decrease in CD3^−^ cells. A low-grade coeliac score >10 was considered positive. 3. Results: Sustained response to GFD was observed in 72 patients (62%) after a median of 21 months of follow-up, who presented more often with coeliac lymphogram (37.5 vs. 11.4%; *p* = 0.02) and a score >10 (32 vs. 11.4%; *p* = 0.027) compared to non-responders. The frequency of low-grade coeliac enteropathy was 19.8%. 4. Conclusion: A GFD is effective in the long-term treatment of patients with previously unexplained chronic watery diarrhoea- or bloating-predominant symptoms fulfilling the criteria of FBD. The response rate was much higher in the subgroup of patients defined by the presence of both a positive low-grade coeliac score and coeliac lymphogram.

## 1. Introduction

Functional bowel disorders (FBDs) are a subset of a larger family of functional gastrointestinal disorders and are associated with chronic symptoms such as abdominal pain, bloating, diarrhoea, and constipation [1]. Similar to other functional disorders, FBDs have no identifiable structural or biochemical abnormalities that can account for their defining symptoms. Diagnosis is, therefore, based on reported symptoms and physical examination, in accordance with the Rome IV criteria, which are the most widely accepted standard for such symptom-based diagnoses. FBDs include, among others, irritable bowel syn drome (IBS), functional diarrhoea, and functional abdominal bloating/distention. In addition, IBS subtypes are defined in Rome IV based on the typical type of stool consistency abnormality (diarrhoea, constipation, and mixed).

Non-coeliac gluten sensitivity (NCGS), which is sometimes referred to as gluten sensitivity, gluten intolerance, or non-coeliac wheat sensitivity, is characterized by intestinal and extra-intestinal symptoms related to the ingestion of gluten-containing food in subjects that are not affected by either coeliac disease (CD) or wheat allergy. This is the original definition based on the Salerno Experts’ Criteria [2]. At present, however, it is recognized that symptoms occur due not just to the ingestion of gluten proteins but potentially other wheat-related components, such as fructans [3,4]. Patients with NCGS have clinical symptoms that are indistinguishable from an IBS-like clinical picture. Conversely, recent studies support the hypothesis that gluten and other wheat components may trigger IBS symptoms [5]. In fact, diet has always played a significant role in IBS, with approximately two thirds of patients developing symptoms soon after the ingestion of food [6,7]. Recent research on diet therapy in IBS has focused on the role of a diet low in fermentable oligo-, di-, and mono-saccharides and polyols (FODMAPs) and wheat-free and gluten-free diets (GFD) [3,5]. Low-FODMAP diets are characterized by the elimination of wheat, barley, spelt, rye, and all other gluten containing cereals, as these cereals also contain fructans, which, as mentioned above, may be responsible for triggering IBS-related symptoms. In fact, a GFD has been proposed as a ‘bottom up’ approach to reducing fructan intake in a low-FODMAP diet [8]. The effect of GFD on patients with symptoms suggestive of IBS has been studied in a number of trials that were limited by small sample sizes and a short study duration, with an overall efficacy ranging from 34 to 71% [9,10]. Double-blind placebo-controlled trials evaluating the role of gluten reintroduction in patients with IBS and symptoms controlled on a GFD have recently been reviewed [3].

It is also known that CD patients may present with IBS-like symptoms. Performing a differential diagnosis between CD and NCGS is sometimes difficult and is especially challenging in cases with low-grade coeliac enteropathy in which CD serology is generally negative. Low-grade coeliac enteropathy lies in the milder range of the CD spectrum and was previously referred to with several different terms, including ‘coeliac-light’, ‘coeliac-lite’, ‘coeliac trait’, ‘mild enteropathy coeliac disease’, and ‘low-grade gluten sensitive enteropathy’ [11,12,13,14,15,16]. We have shown that a blinded gluten challenge in these patients was associated with a significantly higher clinical relapse rate and a deterioration in quality of life as compared with placebo, reinforcing the role of gluten in the pathogenesis of this mild enteropathy [17]. In addition, we derived a scoring system that identifies patients with coeliac characteristics likely to respond to a GFD and to be diagnosed with low-grade coeliac enteropathy with an AUC value of 0.91 [18]. This score uses data on coeliac serology, coeliac genetics (HLA-DQ2/8), and the number of intraepithelial lymphocytes (IEL) and CD3^+^ T-cell receptor gamma-delta^+^ cells (TCRγδ^+^ cells) in duodenal mucosa. In addition, coeliac lymphogram, which is defined as an increase in TCRγδ^+^ IEL plus the additional concomitant decrease in CD3^−^ cells, adds specificity to the IEL assay [18,19,20,21,22]. It has been described that the number of TCRγδ^+^ IELs is only elevated in CD subjects, while in NCGS patients, the number of TCRγδ^+^ IELs is similar to that in controls [23].

The aim of the present study was to evaluate the long-term response rate to a GFD in patients with symptoms suggestive of either diarrhoea- or bloating-predominant FBD and to assess whether or not a low-grade coeliac score value >10 and the presence of coeliac lymphogram increases the probability of response to the diet.

## 2. Materials and Methods

From April 2010 to December 2017, all patients from whom duodenal biopsies were taken to rule out CD were prospectively recorded. The indications for duodenal biopsy sampling were long-standing gastro-intestinal or extra-intestinal symptoms suggestive of CD and/or positive coeliac serology. In addition, most patients were referred for duodenal biopsies on the additional basis of positive HLA-DQ2.5/8.

In the present study, we included consecutive patients recorded in that database based on the following inclusion criteria: (1) age 18 years or over; (2) fulfilling Rome IV criteria of FBD (IBS-D, functional diarrhoea, or functional abdominal bloating); (3) undergoing duodenal biopsies performed while on a gluten-containing diet for both pathology and flow cytometry studies; (4) starting a GFD for FBD symptom control; 5) a follow-up after starting GFD longer than six months to reduce the possibility of a placebo response. Patients were excluded if they had: (1) coeliac disease with atrophy; (2) positive coeliac serology (IgA anti-tissular transglutaminase antibodies—anti-tTG-), even those with anti-tTG borderline titres, defined as those detectable but below the manufacturer cut-off, who had positive IgA anti-endomysial antibodies (EmA); (3) inflammatory bowel disease; (4) microscopic colitis; (5) other enteropathies (olmesartan, giardiasis, etc.).

Demographic data, clinical presentation, coeliac serology (anti-tTG and EmA if indicated), coeliac genetics (HLA-DQ2.5/2.2/8), duodenal histology, IEL count, percentage of TCRγδ^+^ and CD3^−^ cells, and low-grade coeliac score were recorded for all included patients. A retrospective review of the medical records of all these patients to assess the response rate to a GFD was performed. A GFD was administered on the criteria of the physician in charge. Assessment of diet compliance was performed by a dietician when in the 3-month follow-up visit, there was a suspicion of non-adherence following a direct clinical interview with the patient. Afterwards, visits were at 6 months and after that every year during follow-up.

### 2.1. Clinical Response to a GFD

Response to GFD was defined as the sustained complete resolution of symptoms for more than six months, and renewed symptom relapse with inadvertent exposures to gluten-containing foods. In patients with chronic watery diarrhoea, defined as three or more liquid stools per day at least three days in a week, response to a GFD was considered as the complete resolution of diarrhoea. In the case of abdominal bloating, defined as symptoms of bloating and/or distention occurring either daily or at least 3 days a week, being the predominant symptoms in the past 3 months, response to the diet was defined as the sustained complete disappearance of bloating and/or distension. Non-responders to the diet were defined as those with persisting symptoms after a six-week GFD. Partial clinical responses to a GFD were considered as failures considering the retrospective nature of the study and the impossibility to quantitatively measure the response. Therefore, we considered as response only the absence of symptoms, i.e., a clear a meaningful clinical improvement. This response should be maintained at least for 6 months to consider response to a GFD.

### 2.2. Coeliac Serology

Serum IgA anti-tTG (or IgG anti-tTG in IgA deficient patients) was analysed using homologated commercial quantitative automated ELISAs, while the patients were on a gluten-containing diet. As mentioned, patients with anti-tTG titres that were detectable but below the cut-off suggested by the manufacturer were tested for EmA and included only if negative. Serum EmA was performed by indirect immunofluorescence assay in serum samples at 1:5 dilution (commercial sections of monkey distal oesophagus; BioMedical Diagnostics, Marne-la-Vallée, France). Total serum IgA was measured using rate nephelometry (BN II, Siemens Healthcare Diagnostics SL, Marburg, Germany).

### 2.3. Histological Studies

Two endoscopic biopsies from the bulb and four from the second portion of the duodenum were obtained and placed in separate vials in the index endoscopy for standard histological studies while patients were on a gluten-containing diet. Duodenal samples were processed using haematoxylin/eosin staining and CD3 immunophenotyping. Lymphocytic enteritis was considered as an IEL count of >25 IELs per 100 epithelial nuclei and normal villous architecture.

### 2.4. Flow Cytometry

For IEL flow cytometry, one single duodenal biopsy from the second portion of the duodenum was obtained in the index endoscopy and processed immediately as previously described [17,22]. The results of the flow cytometry were obtained in four hours. Coeliac lymphogram was then defined as an increase in TCRγδ^+^ cells >8.5% plus a concomitant decrease in CD3^−^ cells <10%. There were four intraepithelial lymphocyte patterns: a normal pattern, an isolated decrease in CD3^−^, an isolated increase in TCRγδ^+^, and the coeliac lymphogram (an increase in TCRγδ^+^ plus a decrease in CD3^−^). A brief methodological description of the procedures is provided in Appendix B.

### 2.5. Coeliac Genetics

Methods of assessment of coeliac genetics are described in Appendix B.

### 2.6. Low-Grade Coeliac Score and Definition of Low-Grade Coeliac Enteropathy

The low-grade coeliac score was calculated as described previously (Table 1) [18]. We use a cut-off >10 points for positive scores. In the present study, in which all included patients had negative coeliac serology, the score ranged from −2 to 17 points. Low-grade coeliac enteropathy was defined as both a score >10 and a long-term clinical response to a GFD.

The low-grade coeliac score includes among its items the increase in TCRγδ^+^ cells, either isolated or with the concomitant decrease in CD3^−^. In the present study, we analyse the GFD response rate in patients with a positive score comparing both IEL cytometry patterns.

### 2.7. Statistical Analysis

Results are expressed as mean ± SEM and as proportions. Chi-square statistics were used to compare qualitative variables, and either the Student t test or an analysis of variance was used to compare quantitative variables. Statistical calculations were performed using the SPSS for Windows statistical package (SPSS Inc., Chicago, IL, USA). Statistical significance was predetermined as *p* < 0.05. The study SPSS database can be found as Appendix A.

### 2.8. Ethical Issues

The study was conducted in accordance with the Declaration of Helsinki, and the protocol for the prospective registry was approved by the Ethics Committee of the Hospital Universitari MútuaTerrassa at the start of the registry in 2010 (Code: EO/1011; date: 25-03-2010). All participants provided informed consent for that. Since the assessment of GFD response was a retrospective, non-interventional medical record review, informed consent was not requested from patients. Researchers guaranteed strict measures for preserving patient confidentiality. The Ethics and Research Committee of the Hospital Universitari Mútua Terrassa was informed of the conduct of the medical record review.

## 3. Results

During the study period, a duodenal biopsy to rule out CD was performed in 260 patients with FBD, of whom 116 had been treated with a GFD. Eighty-four per cent were HLA-DQ2.5/DQ8/DQ2.2 positive, 44% presented with an IEL coeliac pattern, and 25% presented with a low-grade coeliac score >10. Three (2.6%) patients had detectable anti-tTG titres with negative EmA. As compared to the total sample of 260 patients, the frequency of an IEL count > 25%, an IEL coeliac pattern, and a score >10 was significantly higher in the subsample of patients on a GFD (see Appendix C: Table A1).

### 3.1. Response to Gluten-Free Diet

Clinical response to a GFD was observed in 72 of the 116 patients (62%; 95% CI, 53 to 70%), which was sustained after a median follow-up of 21 months (IQR, 12 to 36). These patients presented more often with the coeliac lymphogram pattern (37.5 vs. 11%; *p* = 0.02) and/or a score >10 (32 vs. 14%; *p* = 0.027) as compared to non-responders (Table 2). Response to GFD increased according to the presence of analytical parameters related to CD. In this sense, patients with a low-grade coeliac score ≤10 had the lowest GFD response rate (55.7%), which progressively increased to 86% in patients with a score >10 and positive coeliac lymphogram (*p* = 0.011) (Figure 1). The response rate to the diet was significantly different in terms of the type of IEL coeliac pattern observed (Figure 1). Those patients presenting with an isolated increase in TCRγδ^+^ cells (*n* = 20) had a response rate of 55%, whereas for those with coeliac lymphogram (*n* = 32), the response rate was 84.4% (*p* = 0.02). In fact, seven out of the 20 (35%) patients with an isolated increase in TCRγδ^+^ cells and 21 out of the 32 (65.6%) patients with coeliac lymphogram had a score >10 (*p* = 0.046).

### 3.2. Frequency of Low-Grade Coeliac Enteropathy

Among the 72 GFD responders, there were 23 patients with a low-grade coeliac score >10 and 49 with a score ≤10. Thus, 23 out of 116 (19.8%) patients were diagnosed with low-grade coeliac enteropathy. Three patients among those with a score ≤10 presented with an HLA-DQ2.5+ and had a low-grade coeliac score equal to 10 points, because they had an IEL count between 19 and 25%, which scores 0 points. We considered that these patients had an inconclusive diagnosis [18]. Besides, 46 out of the 90 remaining patients (51.1%) had a sustained long-term clinical response to a GFD despite a negative score. Table 3 describes the clinical characteristics of these two groups of GFD responders as compared to non-responders. There were no significant differences in demographic variables, type of FBD symptoms, or the presence of HLA-DQ2.5+. Patients with low-grade coeliac enteropathy had significantly higher IEL counts and, as expected by the criteria used for diagnosis, more often coeliac lymphogram and a score >10 than the other groups. Noteworthy, there were no significant differences between non-coeliac GFD responders and non-responders.

## 4. Discussion

The current study presents a large series of patients fulfilling Rome IV criteria for FBD treated with a GFD. The results disclose that 62% of subjects with either diarrhoea or abdominal bloating clinical presentation show long-term clinical response to a GFD. In addition, the data support the acceptability of a GFD, since diet observance was maintained in the long term with sustained improvement. There were no differences in the frequency of HLA-DQ2/8+ between GFD responders and non-responders. However, responders more often present with a positive low-grade coeliac score and/or with coeliac lymphogram. In fact, the response rate of those patients with both a positive score and coeliac lymphogram was 86%, which is significantly higher than the 56% recorded for patients with a negative score.

The low-grade coeliac score was derived statistically to identify patients likely to respond to a GFD and be diagnosed with low-grade coeliac enteropathy with a sensitivity of 86% and a specificity of 85.2% [18]. Sensitivity is lower for patients with negative coeliac serology (77%), maintaining the same specificity (85%). Low-grade coeliac enteropathy is a term that was proposed to describe those patients characterized by lymphocytic enteritis (Marsh 1 enteropathy), positive coeliac genetics, and clinical and histological remission after a GFD [18]. Most of these patients had negative coeliac serology and present with an increased intraepithelial TCRγδ^+^ cells count. As quoted above, several authors have considered that these patients present a mild form of CD [11,12,13,14,15,16], but despite that, they are frequently not treated as coeliacs with a GFD, and this is troubling, since both our own and other previous studies have shown that these patients may present with intestinal and extraintestinal symptoms compatible with the CD clinical spectrum, which improve after a GFD [15,16,24,25,26]. In this setting, the low-grade coeliac score represents a quantitative measure of the ‘coeliac trait’ described by Popp and Mäki [15]. Using dermatitis herpetiformis as a model disease in which there are gluten-related symptoms despite a non-atrophic enteropathy, even with negative coeliac serology in 60% of patients [27], these authors argue about the existence of a ‘coeliac trait’, consisting of a Marsh 1 lesion, positive coeliac genetics, and increase in TCRγδ^+^ cells, which should be identified and treated.

While a high density of TCRγδ^+^ intraepithelial lymphocytes in patients with non-atrophic enteropathy who also carry the susceptibility genes for CD seems to be a prerequisite for developing CD [28,29], this is not pathognomonic for the disease [19,22,29]. The low-grade coeliac score uses the TCRγδ^+^ count, and in seronegative patients, this is the parameter that scores higher. However, results of the present study clearly show that the increase in TCRγδ^+^ cells only has diagnostic value in seronegative Marsh 1 patients if there is a concomitant decrease in CD3^−^ cells, i.e., when coeliac lymphogram is present. In fact, patients with a positive score presented significantly more often with the coeliac lymphogram than with an isolated increase in TCRγδ^+^. Previous studies have shown a higher specificity in CD diagnosis for the coeliac lymphogram than for the isolated increase in TCRγδ^+^ [19,30,31]. Since an isolated increase in TCRγδ^+^ cells is not a useful biomarker of response to a GFD and, thus, of low-grade coeliac enteropathy, methods such as immunohistochemistry, which only measure this parameter, are not useful in this setting. Therefore, coeliac lymphogram assessed by flow cytometry should be used instead, since it allows for the concomitant determination of CD3^−^ cells, thereby increasing the diagnostic accuracy of the assay [18,19,21,22]. Taking an additional duodenal biopsy for flow cytometric analysis can provide useful information for decision making. Most laboratories in tertiary and even secondary hospitals dispose of a flow cytometer for diagnostic purposes, and analysing the lymphocyte subpopulations in the duodenal mucosa is an affordable technique.

Additionally, our results confirm that a cut-off of 25% IEL significantly increases the probability of low-grade coeliac enteropathy. However, as previously shown [18,31], there were a number of patients with lower cut-offs (between 19 and 25%) who were also likely to be diagnosed with low-grade coeliac enteropathy.

The response rate to GFD observed was within the range reported by previous studies. A prospective study of 41 patients with IBS-D showed clinically significant improvements in the IBS symptom severity score after six weeks on a GFD, without significant differences between HLA-DQ2/8-positive and -negative subjects. Twenty-nine of the 41 patients (71%) with clinical response were followed up for 18 months, and 21 were still on a GFD with sustained clinical response [9]. In another study, 12 out of 35 IBS-D or IBS-M patients (34%) clinically improved after a four-month period on a GFD. Additionally, the expression of HLA-DQ2/8 was not useful as diagnostic marker for GFD response [10]. As mentioned above, there are also other studies showing the effect of gluten exposure in IBS-D patients, which have recently been reviewed [3].

Independently of the presence or not of CD tissue biomarkers, the response rate to a GFD was very high in patients with symptoms suggestive of diarrhoea- or abdominal bloating-predominating FBD. In these patients, a GFD may be useful for treating patients with a low-grade coeliac enteropathy, as well as those with NCGS. In this sense, the most probable diagnosis of the non-coeliac GFD responders in the present study was NCGS. A formal diagnosis would require performing a gluten vs. placebo-controlled oral provocation [2]. However, this is controversial as the culprit triggering NCGS is currently unknown [32]. In this sense, results of a recent controlled double-blind crossover challenge study suggest that fructans rather than gluten seem to be the cause of symptoms in patients considering themselves as ‘gluten-sensitive’ [33].

Since a GFD may lead to a reduction in fructan intake that is sufficient to achieve sustained clinical improvement in non-coeliac individuals and may also be effective when treating those with a low-grade coeliac enteropathy, gluten restriction seems to be an effective initial approach for patients presenting with previously unexplained diarrhoea and/or abdominal bloating of presumably functional origin. In fact, it has been suggested that a GFD may be the easiest way of achieving fructan reduction [4], since fructans are a key component to be reduced in a long-term adapted low-FODMAP diet, as demonstrated in a prospective study of 103 patients [34]. In this sense, it has been suggested that a GFD may be administered as a ‘bottom-up’ approach in the FODMAP diet for patients with IBS. This ‘bottom-up’ approach has been advocated as a way to avoid prolonged dietary restrictions in a low-FODMAP diet, potentially avoiding disruption to the gut microbiota and to nutritional status [35]. In addition, patients have rated a GFD as more acceptable than a low-FODMAP diet [36], and only 40% of patients have been shown to follow the low-FODMAP diet correctly [37].

The present study has a number of drawbacks. Firstly, the retrospective nature of the evaluation of dietary response is one limitation of the study; however, we considered response to GFD only if a complete and sustained resolution of symptoms was observed after at least 6 months of follow-up. This fact together with symptom relapse with inadvertent gluten exposure and long-term maintained observance to diet suggest a true response to gluten restriction. Secondly, the study was non-controlled, although a systematic meta-analysis of randomized controlled trials in IBS has demonstrated a pooled placebo response rate of 37.5%, with lower responses seen in those patients who fulfil the Rome criteria on study entry and who received eight weeks or more of therapy [38]. This suggests that in our study, the 62% response rate to a GFD is unlikely to be a placebo effect particularly because improvement was maintained at a median of 21 months. Third, the present study, unlike previous ones, was performed mostly in individuals having positive coeliac genetics (79% HLA-DQ2.5 and/or DQ8+ plus 5% HLA-DQ2.2+). However, this isolated parameter is not a good biomarker of response to a GFD, as has been shown both in several previous studies discussed above and in the present study, probably because of the high prevalence of these genes in the general population. Finally, the frequency of a positive low-grade coeliac score and coeliac lymphogram was higher in the sample of patients treated with a GFD than in the entire sample of 260 patients with FBD. This suggests that the actual rate of low-grade coeliac enteropathy is probably somewhat lower than the observed rate.

In conclusion, a GFD is effective in the long-term treatment of patients with previously unexplained chronic watery diarrhoea- or bloating-predominant symptoms fulfilling the criteria of FBD. The response rate is much higher in a subgroup of patients defined by the presence of both a positive low-grade coeliac score and coeliac lymphogram who may be diagnosed with low-grade coeliac enteropathy. It is mainly the presence of coeliac lymphogram and not the increase in TCRγδ^+^ cells that is useful as a tissue biomarker of low-grade coeliac enteropathy. The results support the recommendation of administering a GFD as a ‘bottom-up’ approach in the FODMAP diet for patients with IBS.

## Figures and Tables

**Figure 1 nutrients-13-01812-f001:**
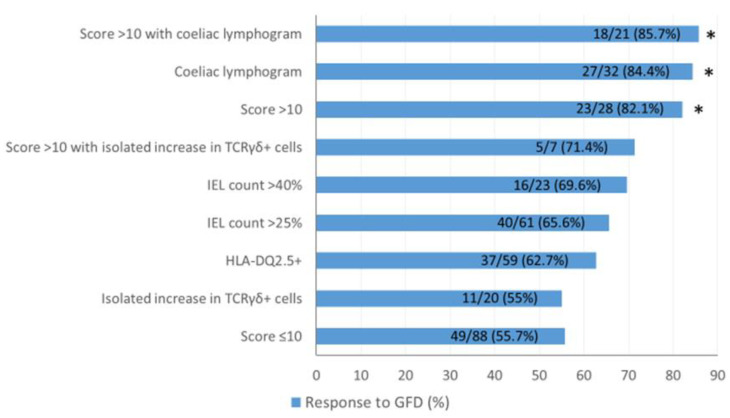
Response rate to a GFD in function of the presence of the different coeliac parameters and the low-grade coeliac score (* *p* < 0.012 vs. score ≤ 10). The different parameters are interrelated, the score integrates individual parameters, and it is not possible to separate the scoring system from the coeliac lymphogram: most patients with a positive score (>10) had a coeliac lymphogram and vice versa.

**Table 1 nutrients-13-01812-t001:** The low-grade coeliac scoring system (−2 to 25 points): a score >10 points is considered positive [18].

Predictors	Points
Serum anti-tTG2	
>20 U/mL	10
>8–20 U/mL or >2–8 U/mL plus EmA+	6
>2 to 8 U/mL plus EmA-	2
2 U/mL	0
IEL cytometry pattern	
↑TCRγδ^+^ cells	7
Histology (IEL count)	
>25%	5
19–25%	0
<19%	−1
Coeliac genetics:	
DQ2.5+	3
DQ8+/DQ2.2+/Allele DQB1 of haplotype DQ2.5+	0
2 alleles DQ2.5- and DQ8-	−1

Serum anti-tTG2: IgA anti-transglutaminase antibodies; EmA: IgA anti-endomysium antibodies; IEL: intraepithelial lymphocytes.

**Table 2 nutrients-13-01812-t002:** Description of patients receiving GFD (*n* = 116): comparison of patients in terms of their response to the diet.

	Total (*n* = 116)	Response (*n* = 72)	Non-Response (*n* = 44)	*p* Value
Type of FBD symptoms:				
-IBS-D or functional diarrhoea -Functional bloating	68 (58.6%) 48 (41.4%)	39 (54.2%) 33 (45.8%)	29 (65.9%) 15 (34.1%)	0.21
Age (mean ± SEM)	42.4 ± 1.24	41.7 ± 1.6	43.7 ± 2	0.44
Sex (% female)	90 (77.6%)	55 (76.4%)	35 (79.5%)	0.69
Coeliac genetics:				
-HLA-DQ2.5 -HLA-DQ8 -HLA-DQ2.2 -1 allele DQ2.5 -Negative	59 (51.8%) 32 (27.6%) 11 (5%) 7 (3.2%) 7 (3.2%)	37 (52.9%) 22 (30.5) 6 (8.3%) 2 (2.8%) 5 (6.9%)	22 (50%) 10 (22.7%) 5 (11.3%) 5 (11.3%) 2 (4.5%)	0.77
Serology:				
-Detectable anti-tTG2 titers (EmA neg)	3 (2.6%)	2 (2.8%)	1 (2.3%)	0.87
Histology (IEL count):				
>25% 19–25% <19%	63 (54.8%) 18 (15.7%) 35 (30.2%)	41 (56.9%) 11 (15.3%) 19 (26.4%)	22 (50%) 7 (15.9%) 15 (34.1%)	0.67
Coeliac IEL cytometry pattern:				
-Non-coeliac -Isolated increase in TCRγδ^+^ cells -Coeliac lymphogram	64 (55.2%) 20 (17.2%) 32 (27.6%)	34 (47.2%) 11 (15.3%) 27 (37.5%)	30 (68.2%) 9 (20.5) 5 (11.4%)	0.019
Low-grade coeliac score > 10	28 (24.1%)	23 (31.9%)	5 (11.4%)	0.027
Score > 10 and coeliac lymphogram	21 (18.1%)	18 (25%)	3 (6.8%)	0.023
Score > 10 and isolated increase in TCRγδ^+^ cells	7 (6.0%)	5 (6.9%)	2 (4.5%)	0.71

FBD: functional bowel disease; anti-tTG2: IgA anti-transglutaminase antibodies; EmA: IgA anti-endomysium antibodies; IEL: intraepithelial lymphocytes.

**Table 3 nutrients-13-01812-t003:** Comparison of the study variables among patients with low-grade coeliac enteropathy (LGCE), functional bowel disease GFD responders (FBD-R), and non-responders (FBD-NR) *.

Variable	LGCE (*n* = 23)	FBD-R (*n* = 46)	FBD-NR (*n* = 44)	*p* Value
Age (years) (mean ± SEM)	44.6 ± 2.8	39.8 ± 1.8	43.7 ± 2.0	0.24
Sex (% women)	15 (65.2%)	38 (82.6%)	35 (79.5%)	0.25
Type of FBD:				
-IBS-D/functional diarrhoea -Functional bloating	12 (52.2%) 11 (47.8%)	26 (56.5%) 20 (43.5%)	29 (65.9%) 15 (34.1%)	0.49
HLA-DQ2.5+	14 (63.6%)	20 (44.4%)	22 (50%)	0.34
LE (IEL > 25%) (%)	23 (100%)	19 (41.3%)	22 (50%)	<0.001
IEL count (mean ± SEM)	38.4 ± 3.4	24.5 ± 1.9	26.1 ± 2.3	0.001
Low-grade coeliac score >10	23 (100%)	0	5 (11.4%)	<0.001
Low-grade coeliac score (mean ± SEM)	13.9 ± 0.4	4.7 ± 0.5	5.7 ± 0.4	<0.001
Coeliac IEL cytometry pattern:				
-Isolated increase in TCRγδ^+^ cells -Coeliac lymphogram	5 (21.7%) 18 (78.3%)	6 (14%) 7 (15.2%)	9 (20.5%) 5 (11.4%)	<0.001
TCRγδ^+^ cells (%) (mean ± SEM)	20.6 ± 2.3	7.7 ± 1.4	6.9 ± 1.1	<0.001
CD3^−^ cells (%) (mean ± SEM)	6.6 ± 1.1	15.9 ± 1.9	18.3 ± 1.9	<0.001

LE, lymphocytic enteritis; IEL: intraepithelial lymphocyte; * Three patients with a response to the GFD were excluded from this evaluation, since it was not possible to differentiate between LGCE and FBD-R. Two of them had coeliac lymphogram, and one had an isolated increase in TCRγδ^+^ cells (see text).

## Data Availability

Database of the study supporting the reported results can be found as a Appendix A.

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
