# Peer review of "Long-Term Effect of a Gluten-Free Diet on Diarrhoea- or Bloating-Predominant Functional Bowel Disease: Role of the ‘Low-Grade Coeliac Score’ and the ‘Coeliac Lymphogram’ in the Response Rate to the Diet"

_nutrients, 2021, doi:10.3390/nu13061812_

Round 1

Reviewer 1 Report

In the present study Fernandez-Banares et al demonstrated that gluten free diet (GFD) is beneficial in patients with functional disorders (FD), in particular diarrhea and abdominal bloating. Furthermore, the subgroup with a typical celiac lymphogram and high celiac score had a better chance to benefit from GFD. Main comments:

1) In the Abstract, please do not be generic and state that IBS-D, functional diarrhea and functional bloating have been enrolled, as reported in lines 92-93.

2) The role of IELs in the pathogenesis of FD should be briefly discussed (see Losurdo G et al, Cell Mol Immunol 2021).

3) Results should be reported using SD, and not SEM.

4) What about smoke? Indeed it has been found that it is related with more intense IEL infiltration in the duodenum of IBS patients (see Losurdo G et al, Medicina 2020). Please comment.

5) I do not understand why some patients (line 218) were classified as low grade celiac enteropathy if CD was ruled out. This term is misleading in this context and should be avoided. Instead, microscopic enteritis or duodenal lymphocytosis are preferable (Rostami K et al, World J Gastroenterol 2015; Ierardi E et al, Ann Gastroenterol 2017).

Author Response

REVIEWER 1

In the present study Fernandez-Banares et al demonstrated that gluten free diet (GFD) is beneficial in patients with functional disorders (FD), in particular diarrhea and abdominal bloating. Furthermore, the subgroup with a typical celiac lymphogram and high celiac score had a better chance to benefit from GFD. Main comments:

1) In the Abstract, please do not be generic and state that IBS-D, functional diarrhea and functional bloating have been enrolled, as reported in lines 92-93.

Response: We mention that 116 adult patients with either predominant diarrhea or abdominal bloating were treated with a GFD. We have added: fulfilling Rome IV criteria of FBD (line 16).

2) The role of IELs in the pathogenesis of FD should be briefly discussed (see Losurdo G et al, Cell Mol Immunol 2021).

Response: Lymphocytic duodenosis is an unspecific finding, many causes have been suggested, and even it may be idiopathic (Rosinach M et al, Dig Liver Dis 2012; Aziz et al. Aliment Pharmacol Ther 2010). However, the presence of an increase in CD3+TCRgammadelta+ IEL has been suggested to be a feature of coeliac disease, and it is not observed in patients with non-coeliac gluten sensitivity (this is explained in the introduction of the paper with the appropriate references). In the present study, we evaluated patients with symptoms suggestive of FBD (lines 83, 304, 317, 348), but as in other clinical situations (p.e., bile acid diarrhoea, microscopic colitis, some coeliac disease patients) an organic cause may exist to explain the symptoms. After being diagnosed with ‘low grade coeliac enteropathy’ or ‘non-coeliac gluten/wheat sensitivity’, patients had a diagnosis different from FBD, but the initial symptoms are similar. Thus, we think that to comment the role of IELs in the pathogenesis of FBD is beyond of the scope of the present paper.

3) Results should be reported using SD, and not SEM.

Response: Standard error measures the precision of the estimate of the sample mean. If we need to draw conclusions about the spread and variability of the data, standard deviation is recommended. However, if we are interested in finding how precise the sample mean is or we are testing the differences between two means, standard error is the preferred metric. In our paper we give mean±SEM since we are comparing means between groups.

4) What about smoke? Indeed it has been found that it is related with more intense IEL infiltration in the duodenum of IBS patients (see Losurdo G et al, Medicina 2020). Please comment.

Response. We have read with interest the results of the paper of Losurdo et al in which smoking was related to the IEL infiltration. Regrettably, this was based on only 7 smoking patients. This low sample size precludes to draw solid conclusions. In our study we have no data about smoking habit. However, and as mentioned in the answer to question 2, IEL infiltration is in general unspecific and smoking could be an additional aetiological factor, but it is likely not related to the increase in TCRgammadelta+ cells.

5) I do not understand why some patients (line 218) were classified as low grade celiac enteropathy if CD was ruled out. This term is misleading in this context and should be avoided. Instead, microscopic enteritis or duodenal lymphocytosis are preferable (Rostami K et al, World J Gastroenterol 2015; Ierardi E et al, Ann Gastroenterol 2017).

Response: As mentioned in the introduction section, the term ‘Low-grade coeliac enteropathy’ refers to patients with lymphocytic duodenosis who present with characteristics of coeliac disease (i.e., the ‘celiac trait’ described by Popp and Mäki. Nutrients 2019), and lies in the milder range of the CD spectrum and was previously referred to with several different terms, including ‘coeliac-light’, ‘coeliac-lite’, ‘mild enteropathy coeliac disease’ and ‘low-grade gluten sensitive enteropathy’ [references, 11 to 16 of the paper]. Classical CD with villous atrophy was ruled out at inclusion, but to detect this mild coeliac enteropathy defined by duodenal lymphocytosis, HLA-DQ2.5+, increase in TCRgammadelta+ cells (with the concomitant decrease in CD3neg cells: named ‘coeliac lymphogram’), and coeliac serology either borderline or negative, among the patients with response to a GFD was one of the aims of the present study.

As commented in the discussion this mild enteropathy is often the only intestinal damage seen in dermatitis herpetiformis, which is clearly a manifestation of coeliac disease.

Reviewer 2 Report

This paper is not meaningfully different from its first submission.   To assess a) the effect of GFD in FBD patients and b) the role of two composite parameters to predict the response to GFD the authors retrospectively studied 116 patients who had a duodenal biopsy taken and had been treated with a GFD for more than six months.   Major problems: I fear the authors themselves are right to note that treatment efficacy cannot be reliably determined in this uncontrolled, retrospective study.   Other points: a) The classification of responders/non-responders to GFD probably needs clarification: Non-responders were without response after six weeks GFD, what kind of responses were assessed when, how, and what thresholds were used? An absence of symptoms for more than six months in responders is difficult to reconcile with the diagnosis of FBD, again, how was this as welll as  symptom relapse after gluten exposure determined? b) Currently it appears often unclear whether the response to GFD is considered as  dependent or independent variable; e.g. Table 3 seems, apart from age and sex distribution, largely tautologic in its demonstration that the parameters used to define LGCE are indeed more frequent in LGCE. c) Tables: the centered leading column with non-matching numbers of lines in the data columns are difficult to read

Author Response

REVIEWER 2:

This paper is not meaningfully different from its first submission.   To assess a) the effect of GFD in FBD patients and b) the role of two composite parameters to predict the response to GFD the authors retrospectively studied 116 patients who had a duodenal biopsy taken and had been treated with a GFD for more than six months.  

Major problems: I fear the authors themselves are right to note that treatment efficacy cannot be reliably determined in this uncontrolled, retrospective study.   Other points: a) The classification of responders/non-responders to GFD probably needs clarification: Non-responders were without response after six weeks GFD, what kind of responses were assessed when, how, and what thresholds were used? An absence of symptoms for more than six months in responders is difficult to reconcile with the diagnosis of FBD, again, how was this as welll as  symptom relapse after gluten exposure determined?

Response: We agree with the referee about we did not use any scale to evaluate the clinical response in our patients due to the retrospective nature of the study. As we mention (line 113), in patients with chronic watery diarrhoea, defined as three or more liquid stools per day at least three days in a week, response to a GFD was considered as the complete resolution of diarrhoea. This is an objective parameter since number of daily stools and consistency are registered in the medical record of the patient, this corresponds to 59% of evaluated patients. Abdominal bloating was the main clinical symptom in 41% of the patients, which occurred more than 3 days a week (lines 115-116). After a GFD, these patients were symptom-free in the follow-up visits, only explaining symptom recurrence after gluten exposition.

Patients with non-response after six weeks on a GFD, are those in whom symptoms persisted after this period. We consider as response only the absence of symptoms, i.e., a clear a meaningful clinical improvement. This response should be maintained at least for 6 months to consider response to a GFD (this has been added, lines 121 to 124).

We have included patients with symptoms suggestive of FBD (as stated in lines 304, 317, 348) but as in other clinical situations (p.e., bile acid diarrhoea, microscopic colitis, some patients with seropositive celiac disease with atrophy) an organic aetiology may be the cause of these previously unexplained ‘functional’ symptoms. After being diagnosed with ‘low grade coeliac enteropathy’ or ‘non-coeliac gluten/wheat sensitivity’, patients have a diagnosis different from FBD, but the initial symptoms are similar. In the discussion and conclusion of the paper we have mentioned:  ‘a GFD is effective in the long-term treatment of patients with previously unexplained diarrhoea- or bloating- predominant symptoms’, to better clarify this point (line 351).

  1. b) Currently it appears often unclear whether the response to GFD is considered as dependent or independent variable; e.g. Table 3 seems, apart from age and sex distribution, largely tautologic in its demonstration that the parameters used to define LGCE are indeed more frequent in LGCE.

Response: Among the patients with GFD response we have evaluated the presence of coeliac characteristics (the ‘celiac trait’ described by Popp and Mäki, Nutrientes 2019). The low-grade coeliac score is a statistically derived quantitative approach to measure this ‘coeliac trait’ and to assess the probability of response to a GFD. When a response to GFD is observed after a positive score the patient can be diagnosed with low-grade coeliac enteropathy. Obviously, these characteristics are significantly more frequent in patients considered as ‘low-grade coeliac enteropathy’ than in the other groups. Table 3 mainly shows that there were no differences between ‘non-coeliac’ GFD responders and GFD non-responders. This was remarked in the paper (line 227-228).

  1. c) Tables: the centered leading column with non-matching numbers of lines in the data columns are difficult to read.

Response: I don't kunderstand what the reviewer is referring to. In the tables there are horizontal lines that correctly separate the variables and there are no numbers that do not match.

Round 2

Reviewer 1 Report

Regarding points 1-4, answers are ok

Regarding point 5, I still believe that LGCE is a misleading term, therefore Authors must prefer microscopic enteritis or duodenal lymphocytosis and comment.

Reviewer 2 Report

I do not believe the manuscript has been significantly
improved and now warrants publication in Nutrients